# Transformer for Gene Expression Modeling (T-GEM): An Interpretable Deep Learning Model for Gene Expression-Based Phenotype Predictions

**DOI:** 10.3390/cancers14194763

**Published:** 2022-09-29

**Authors:** Ting-He Zhang, Md Musaddaqul Hasib, Yu-Chiao Chiu, Zhi-Feng Han, Yu-Fang Jin, Mario Flores, Yidong Chen, Yufei Huang

**Affiliations:** 1Department of Electrical and Computer Engineering, The University of Texas at San Antonio, San Antonio, TX 78249, USA; 2Department of Medicine, School of Medicine, University of Pittsburgh, Pittsburgh, PA 15232, USA; 3UPMC Hillman Cancer Center, University of Pittsburgh, Pittsburgh, PA 15232, USA; 4Greehey Children’s Cancer Research Institute, The University of Texas Health Science Center at San Antonio, San Antonio, TX 78229, USA; 5Department of Population Health Sciences, The University of Texas Health Science Center at San Antonio, San Antonio, TX 78229, USA

**Keywords:** phenotypes prediction, interpretable deep learning, Transformer, cancer type prediction, immune cell type prediction

## Abstract

**Simple Summary:**

Cancer is the second leading cause of death worldwide. Predicting phenotype and understanding makers that define the phenotype are important tasks. We propose an interpretable deep learning model called T-GEM that can predict cancer-related phenotype prediction and reveal phenotype-related biological functions and marker genes. We demonstrated the capability of T-GEM on cancer type prediction using TGCA data and immune cell type identification using scRNA-seq data. The code and detailed documents are provided to facilitate easy implementation of the model in other studies.

**Abstract:**

Deep learning has been applied in precision oncology to address a variety of gene expression-based phenotype predictions. However, gene expression data’s unique characteristics challenge the computer vision-inspired design of popular Deep Learning (DL) models such as Convolutional Neural Network (CNN) and ask for the need to develop interpretable DL models tailored for transcriptomics study. To address the current challenges in developing an interpretable DL model for modeling gene expression data, we propose a novel interpretable deep learning architecture called T-GEM, or Transformer for Gene Expression Modeling. We provided the detailed T-GEM model for modeling gene–gene interactions and demonstrated its utility for gene expression-based predictions of cancer-related phenotypes, including cancer type prediction and immune cell type classification. We carefully analyzed the learning mechanism of T-GEM and showed that the first layer has broader attention while higher layers focus more on phenotype-related genes. We also showed that T-GEM’s self-attention could capture important biological functions associated with the predicted phenotypes. We further devised a method to extract the regulatory network that T-GEM learns by exploiting the attributions of self-attention weights for classifications and showed that the network hub genes were likely markers for the predicted phenotypes.

## 1. Introduction

The ever-growing genomics data from omics profiling and single-cell technologies have spurred rapid developments of deep learning (DL) methods for genomics, especially gene expression-based phenotype predictions [1,2]. In precision oncology, deep learning modeling of gene expression and other genomics data have offered exciting new solutions and perspectives to a variety of important questions cancer diagnosis using convolutional neural networks (CNNs) [3,4,5], patient survival prediction using graph convolution networks (GCNs) [6], drug response prediction using deep neural networks (DNN) [7,8], and cancer vulnerability prediction using DNN [9]. While producing accurate predictions is critical, developing deep interpretable DL models capable of informing the functions and mechanisms underlying the phenotype predictions becomes increasingly desirable [10]. 

However, developing interpretable DL models for gene expression-based phenotype prediction faces the following challenges. First, how to model unordered genes: A gene expression sample is commonly represented as a vector of unordered expression values. However, popular CNN was originally developed for images/texts, where features (pixels/words) have a fixed order. Therefore, different gene orders for the same sample could lead to distinct DL architectures with varying performances. Finding the best order is computationally inhibitive. Although DNN is order-agnostic, it is highly inefficient and prone to overfitting. Second, how to model gene–gene interactions: Functional gene interactions impose correlations between their expressions. However, interacting genes might be separated in the expression vector. Because CNNs exploit pixel/text correlations within a local region, they cannot fully capture functional interactions from the expression data. Existing attempts to map gene expression data into 2- or 3-D images are inadequate to model functionally interacting genes [11,12,13]. The recent development of graph CNN (GCNN) points to a promising direction for addressing this issue [6]. However, GCNN is restricted to working with a predefined network and cannot model context-specific interactions. Third, inferring functions and markers from (G)CNN/DNN is not trivial. As the convolution/dense layer bears little biological considerations, its computation muddles input genes’ identities, generating feature maps highly difficult to deconvolute. The disagreement between functional interactions and image local features hinders the existing DL interpretation tools from extracting much-desired insights into regulatory pathways and functions. Moreover, phenotypes can be defined by either a few markers or some complex signaling cascades. Because neighboring dependencies in image/text data encourage “smoothness,” the objective of revealing isolated markers or networks is at odds with the smooth constraint of many DL interpretation algorithms. Therefore, genomics data’s unique characteristics challenge the computer vision-inspired design of popular DL models such as CNN and ask for the need to develop interpretable DL models tailored for transcriptomics study.

One notable class of genomics-centric DL models is visible neural networks (VNNs) [14,15], which include models such as DCells [16], DrugCells [17], and P-NET [15]. They are DNN models whose connections and hierarchical layers are defined based on Gene Ontology (GO), pathways, or networks. They have been trained to predict gene dependencies in yeast and drug responses in cancer cells, where the model architectures could inform the associated functions [16,17]. Although VNNs facilitate interpretation, they are inflexible to model and uncover context-specific interactions as the connections and hierarchy are fixed. Moreover, they are oversized models for my genomics applications where the sample size is small. For instance, DCells and DrugCells have 12 layers but only model binary perturbations or mutations.

To address the above limitations, we propose a new interpretable DL modeling framework based on Transformer. Since their invention in 2017 [18], Transformer has taken natural language processing (NLP) by storm. The Transformer and the Transformer-based models have quickly achieved state-of-the-art performance for most NLP tasks [18,19]. The Transformer’s core is the self-attention mechanism, which learns global features’ dependencies. We recognize that self-attention addresses the existing DL models’ limitations to model genomics data. Specifically, unlike CNN, self-attention can model unordered input, such as genes in gene expression data. Furthermore, self-attention models’ interactions between all the features instead of local features, as in CNN, thus capable of learning important gene–gene interactions. Above all, self-attention facilitates interpretation. Unlike CNN or DNN, outputs of self-attention take the same input word identity but learn new representations of input words. This property has been exploited to explain that BERT learns the classical NLP pipeline in an explainable manner, encoding different syntactic and semantic forms [20,21,22]. These appealing features and benefits of self-attention inspired us to propose Transformer for Gene Expression Modeling or T-GEM.

In this paper, we hypothesize that: T-GEM can accurately predict phenotypes based on gene expression;T-GEM captures the biological function and potential marker genes.

To assess these hypotheses, we design the T-GEM model for modeling gene–gene interactions and evaluate its ability for gene expression-based predictions and functional interpretation of cancer-related phenotypes, including cancer type prediction and immune cell type classification. The remaining paper is organized as follows. In the Materials and Methods, we describe the datasets and preprocessing steps. We also detail the proposed T-GEM model and discuss the developed methods to interpret T-GEM for its learned functions, key regulatory networks, and marker genes. In the Results, we carefully assess T-GEM’s prediction performance and compare it with existing popular approaches. We further analyze the learning mechanism of T-GEM and show that T-GEM’s self-attention could capture important biological functions associated with the predicted phenotypes. We also extract the regulatory network that T-GEM learns by exploiting the attributions of self-attention weights for classifications and show that the network hub genes were likely markers for the predicted phenotypes. Concluding remarks are discussed in the Conclusion.

## 2. Materials and Methods

In this study, we proposed a novel interpretable deep learning model called T-GEM, or the Transformer for Gene Expression Modeling, to predict gene expression-based cancer-related phenotypes, including cancer type prediction and immune cell type classification. The interpretability of T-GEM is also presented.

### 2.1. Data Collection and Preprocessing

The T-GEM is evaluated on two gene expression-based datasets from RNA-seq and scRNA-seq, respectively.

#### 2.1.1. TCGA RNA-Seq Data

The Cancer Genome Atlas (TCGA) [23] pan-cancer dataset was collected via R/Bioconductor package TCGAbiolinks [24] in December 2018. The dataset contains 10,340 and 713 samples for 33 cancer types and 23 normal tissues, respectively. All gene expression was processed by log-transferred and normalized by min-max. To facilitate the investigation of T-GEM’s learning mechanism and functional interpretation, we selected 1708 genes from 24 genesets containing 14 cancer functional genesets from CancerSEA [25] and 10 KEGG biological pathways that are less relevant to cancer.

#### 2.1.2. PBMC scRNA-Seq Data

To test T-GEM’s generalizability and interpretability, we applied T-GEM for immune call type classification using the single-cell RNA-seq (scRNA-seq) gene expression profile of Human peripheral blood mononuclear cells (PBMC; Zhengsorted) [26]. The PBMC dataset includes 10 cell populations extracted through antibody-based bead enrichment and FACS sorting. Each cell population has 2000 cells and, therefore, there is a total of 20,000 cells from 10 cell classes. We followed the method in [27] to preprocess the data and selected 1000 highly variable genes as input features.

### 2.2. T-GEM Model

Let x∈ℝG×1 be the vector of normalized log-transformed gene expression of G genes and y∈ℤ be the classification label. The goal is to design a deep learning model F parameterized by θ such that p(y|x)=Fθ(x). The overall architecture of T-GEM follows that of a Transformer (Figure 1A), which includes multiple attention layers, each of which is composed of multiple self-attention heads. Similar to the classical Transformer, the attention layer generates new attention values z∈ℝG×1, which are representations of the input gene expression x. One important feature of the attention layer is that there is a one-to-one correspondence between the input gene and the output representation, i.e., zg is the representation of xg (Figure 1B). This input–out consistency is a key modeling feature of T-GEM that results in a biological-aware design of the self-attention units and an essential ingredient that enables an interpretable deep learning model. The core computing element in T-GEM is the self-attention module (Figure 1B), which takes the representation of each gene from the previous layer and computes a set of new representations. We use the self-attention module of the first attention layer to explain its inner operations (Figure 1B). To help understand the connections and differences between the self-attention modules of T-GEM and Transformer, we draw the parallel between an input gene for T-GEM and an input word for Transformer. However, a notable difference lies in the fact that a gene denotes its scalar expression value, whereas a word is embedded by a word vector. Similar to the Transformer, the self-attention module of T-GEM performs two sets of operations. First, three entities, i.e., Query (qg), Key (kg), and Value (vg) are computed for each gene g, where qg=wqgxg, kg=wkgxg, and vg=wvgxg, with wqg, wkg, and wvg being the weights to be learned. Compared to Transformer, they are similar in that all the computations are linear; they are nevertheless different because for Transformer, the linear weights are shared for all the words, whereas for T-GEM, the weights wqg, wkg and wvg are gene-dependent. Introducing gene-dependent weights makes T-GEM more flexible and powerful to learn gene-specific representations. It is also computationally feasible because the input gene is a scalar as opposed to a word vector. In the second operation, the representations for each Query gene g are computed. Specifically, for Query gene g, an attention weight vector ag or a probability distribution of the similarity between g and all other Key genes except g is computed as
(1)agi=softmax(f(qg,ki))    ∀ i=1, …, G and i≠g
where agi is the gth attention weight in ag and f is a similarity between qg and ki defined as
(2)f(qg,ki)=qgki

Equation (2) makes Query attend to the Key gene that has a similar up/down behavior and consistently large magnitudes (either highly or lowly expressed) across different samples. Since the Key genes with such behavior are likely to be marker genes, Equation (2) would result in Query gene paying attention to marker genes. 

Once the attention weights ag are computed, a representation of the Query gene is obtained as
(3)zg=ag⊤vg− 
where vg− is the vector of the values excluding gene g. The computation in Equation (3) suggests that the genes that are deemed more “similar” by Equation (2) (i.e., correlated marker genes) will have bigger contributions to gene g’s representation. This progress is repeated for each gene until the representations for all the genes are computed. This concludes the operations for the self-attention module. When there are H self-attention heads, the representations from each head will be linearly combined to obtain a single set of representations z1 for layer 1
(4)z1=x+layernorm( wH⊤Z) 
where the summation represents the skip connection, wH∈ℝH×1 is the weight vector, Z∈ℝH×G has its (h,g)th element as the representation of the query gene in head h, and layernorm represents the layer normalization function [18]. The entire computation from Equaions (1)–(4) will repeat to calculate the Query gene representations of additional layers. In the end, a classification layer is added
(5)p(y|z)=softmax(ϕ(zL)) 
where ϕ is an activation function and zL is the representations of last layer L.

### 2.3. T-GEM Interpretation Methods

#### 2.3.1. Entropy of the Attention Weights of a Head

The attention weight vector $a_{g}$ measures the attentions that the Query gene g has on the Key genes, and it is in the form of a probability distribution. Therefore, to assess the characteristics of the attentions of a Query gene, we compute the entropy of $a_{g}$
(6)$$H\left(a_{g}\right)=-\overset{G}{\underset{i}{\sum }}a_{gi}\log\left(a_{gi}\right)$$

Intuitively, a lower entropy demonstrates that the distribution of $a_{gi}$ becomes more skewed, and therefore Query gene g pays higher attention to specific genes or focused attention. In contrast, higher entropy indicates that the distribution of $a_{gi}$ is flatter, and therefore Query gene g pays similar attention to its Key genes or broad attention. We averaged the entropies across all samples to obtain the entropy of a Query gene.

#### 2.3.2. Attribution Scores of Attention Weights of a Head and a Layer

To assess the contributions of the attention weights to the classification, we adopted the method proposed in [28] to compute the attribution scores of the attention weights using an Integrated gradient (IG). Given the T-GEM model with attention weights $a_{gi}$ and some baseline weights $a_{gi}{}^{\prime }$, an attribution score is computed as
(7)$$IG\left(a_{gi}\right)=\left(a_{gi}-a_{gi}{}^{\prime }\right)\times \int^{\;}\frac{\partial F\left(a_{gi}+\alpha \times \left(a_{gi}-a_{gi}{}^{\prime }\right)\right)}{\partial a_{gi}}d\alpha$$
where the integral is taken along a straight line from the $a_{gi}{}^{\prime }$ to $a_{gi}$ parameterized by the parameter α. To simply the computation, we set baseline $a_{gi}{}^{\prime }$ as 0. We used the PyTorch interpretability library Captum [29] to compute the attribution scores in Equation (7).

#### 2.3.3. Visualization of the Regulatory Networks of the Top Layer 

A regulatory network representing the regulations of Query genes by Key genes was derived from the last layer using attention weight attribution scores computed from the testing samples of a specific label. We used Cytoscape [30] to visualize this network.

### 2.4. Training Procedure 

The model was trained with optimizer Adam [31], with a batch size of 16 and an epoch of 50. The negative log-likelihood function was used as the loss function. The model was trained on multiple sets of hyper-parameters (Table 1), and the best model parameters were selected based on their validation performance. All of the baseline models were trained according to the same procedure. For CNN, AutoKeras [32] was applied to determine the best model architecture. All of the model training was conducted by using Pytorch 1.9.0 on NVIDIA A100 GPUs.

### 2.5. Performance Evaluation Metrics

The proposed model was evaluated on two multi-labeled datasets while TCGA has the unbalance samples for each class and PBMC has the equal size for each class. We adopted two metrics to assess the models’ performance, including ACC (accuracy), MCC (Matthews correlation coefficient), and AUC (the area under the ROC curve). MCC is a reliable metric that produces a high score only if the model performs well in TP, TN, FP, and FN. ACC and MCC is defined as
(8)$$ACC=\frac{TP+TN}{TP+TN+FP+FN}$$
(9)$$MCC=\frac{TP\cdot TN-FP\cdot FN}{\sqrt{\left(TP+FP\;\right)\cdot \left(TP+FN\right)\cdot \left(TN+FP\right)\cdot \left(TN+FN\right)}}$$
where TP, TN, FP, and FN are the numbers of true positives, true negatives, false positives, and false negatives, respectively. The ROC curve is plotted by using FPR (false positive rate) against TPR (true positive rate)
(10)$$FPR=\frac{FP}{TN+FP}$$
(11)$$TPR=\frac{TP}{TP+FN}\;$$
and AUC was computed as the area under the ROC curve.

## 3. Results and Discussion

In this section, we first compare T-GEM’s performance with CNN and several classical classifiers on the TCGA and PBMC datasets. Then, we analyze the learning mechanism of T-GEM and examine the biological functions and marker genes captured by T-GEM.

### 3.1. T-GEM’s Performance for Cancer Type Classification Using TCGA Data

We evaluate the performance of the T-GEM model on cancer type prediction using TCGA pan-cancer gene expression data. The data contain 10,340 patients with 33 different cancer types and 713 normal samples. The classification includes 34 labels (33 cancer types and normal). For each label, we extracted 70% of the samples as the training set, 10% as the validation set, and 20% as the testing set. To facilitate the evaluation of interpretation, we selected genes from 14 cancer functional gene sets from CancerSEA, including apoptosis, angiogenesis, invasion, etc., and 10 KEGG biological pathways gene sets that are not related to cancer. A total of 1708 genes are used as input features (Detailed information about 14 functional gene sets and 10 KEGG pathways can be found in the Appendix A). Because we know which genes belong to cancer-related genesets, we could examine the genes that each T-GEM’s attention layers focus on and understand if T-GEM’s predictions are based on cancer-related genes and functions and what they are.

To determine the architecture and hyperparameter, T-GEM with multiple sets of hyper-parameters (Table 1) were trained, and the obtained model with the best validation performance included three layers, five heads, and no activation in the classification layer of Equation (5).

We evaluated the ACC, MCC, and AUC of T-GEM, CNN, and several classical classifiers (Table 2), where CNN was trained using AutoKeras so that its architecture and hyperparameters were optimized. As we can see, deep learning-based methods, including T-GEM and CNN, have overall better performance, whereas T-GEM achieves the overall best performance, although the improvement over CNN is small, especially on AUCs. To gain a better understanding of the differences in improvements by T-GEM shown by different performance metrics, we examined the confusion matrices of their predictions. We observed that most of the misclassifications were associated with two cancer types, namely colon adenocarcinoma (COAD) and rectum adenocarcinoma (READ). For SVM, it misclassified all 34 READ samples to COAD, while CNN misclassified about half of the 96 COAD samples to READ. In contrast, T-GEM correctly predicted 84/96 COAD samples and 15/34 READ samples. The lower misclassification by T-GEM was clearly revealed by ACC, but this could only register a slight AUC improvement because of the small sample sizes of these two cancer types. Collectively, T-GEM outperformed CNN and classical machine learning algorithms and could achieve lower misclassifications for especially cancer types with small samples.

#### 3.1.1. Investigation of T-GEM’s Learning Mechanism

Next, we investigated how T-GEM learns to produce a classification prediction. Specially, we sought to understand the importance of each layer/head for classification, what T-GEM focuses on in each layer, and the biological functions of each layer. We designed three following analyses to address these questions.

To assess the importance of a layer/head, we first pruned that layer/head and checked the accuracies after pruning. Note that we consider the head output before the skip connection in this experiment. The larger the accuracy reduction that the pruning causes, the more important the head/layer is. To prune a head, we set this head’s corresponding weight in $w_{H}$ in Equation (4) as 0 while keeping the rest of elements in $w_{H}$ unchanged. Similarly, to prune a layer, we zeroed the weights of all the heads in that layer. The original test samples were used to compute this pruned model’s accuracy.

As can be seen in Table 3, the accuracy after pruning layer 1 drops the most from the original 94.92% to 57.48%, indicating that layer 1 is the essential layer. Examining different heads in layer 1 reveals that head 1 could be the most important head. Intriguingly, we observe much smaller accuracy drops for the heads than the layer. This could be due to heads in the same layer capturing similar features. Notice that there are slight drops associated with layers 2 and 3. Recall that each layer includes a skip connection, and thus, this result suggests that being able to pass the learned information from layer 1 to layers 2 and 3 could be an important mechanism to achieve the overall performance.

To further support this observation, we directly assessed the classification performance of each head/layer. To that end, we took the output from a layer/head and trained an SVM. The ACCs are shown in (Table 4). Now, the larger the accuracy, the more important a head/layer is. Overall, all accuracies of all layers/heads are close to the original, indicating that each layer/head contains a large amount of information needed for accurate classification. Note that the accuracies decrease from layer 1 to layer 3, which supports the conclusion from the pruning experiment that passing outputs of layer 1 to other layers through skip connection ensures achieving the reported performance of the T-GEM model. We also notice that the accuracies of different heads, especially in layer 3 vary. This suggests that features extracted from these heads could be different and different heads might focus on learning information from different genes.

Motivated by this observation, we then investigated what T-GEM focuses on in each layer. Recall from Equation (3), that the head output for each Query is computed as a linear combination of the Values weighted by attention weight vector $a_{g}$. As an important part of T-GEM, $a_{g}$ reflects the attention in the form of the discrete probability distribution that Query gene g pays to all Key genes. When this distribution is flat and close to the uniform distribution, the Query gene pays broad attention to Key genes. Otherwise, the Query gene put more attention on a subset of genes. This distribution varies in each layer and each head depending on what and how Query genes attend to Key genes. To understand if a head pays more attention on a subset of genes (i.e., smaller entropy) or has broad attention (i.e., large entropy), we computed the entropy of $a_{g}$ for every head in each layer averaged across all of the samples (Figure 2). We notice that the entropies of all the heads in layer 1 are similar and among the largest, suggesting that Query genes in layer 1 tend to pay similar and broad attention to key genes. In contrast, layer 2 and layer 3 contain heads with Query genes having lower entropies and layer 3 has some of the genes with the lowest average entropy. This suggests that layers 2 and 3 start to pay more attention to specific genes. To further understand what the specific genes the heads focus on, we examined five Query genes with the lowest entropy in layer 3 (Figure 2). We found that all of them are related to cancer. EMX2 is the candidate tumor suppressor that regulates tumor progression [33]. EPCAM is a well-known marker for tumor-initiating cancer stem cells, and it is a penitential target for cancer therapy [34]. POU5F1 is a penitential prognostic and diagnostic biomarker for various epithelial cancers [35]. A lack of BHMT impacts susceptibility to fatty liver and hepatocellular carcinoma [36]. Finally, GFAP is a classical marker of astrocytoma, and it is marker of less malignant and more differentiated astrocytoma [37].

In summary, T-GEM pays broad attention in layer 1 but has more concentrated attention in higher layers. The genes with more focused attention or lowest entropies in layer 3 are cancer-related genes.

#### 3.1.2. T-GEM Makes Decisions by Focusing on Important Cancer Pathways

Next, we investigated the biological functions that T-GEM learns in each layer. To this end, we assessed the attribution of attention weights to the classification by computing their attribution scores using IG, as in Equation (7). Then, we averaged the weight attribution scores of the same Query and Key gene pairs from all heads in the same layer to obtain the attribution scores of the layer function. Then, according to [28], we thresholded the attribution scores to retain the Query and Key genes associated with the high weight attribution scores. A snapshot of layer 3 is shown in Figure 3A. We see that Query genes VCAN, MMP11, and FBN1 are connected with Key genes, indicating that the weight attribution scores between these Query and Key genes are large and, therefore, the information passed from these Key genes to the Query genes contributes to the classification. Consequently, we called these Query genes informative genes. On the contrary, we called Query genes with no linked Key genes (e.g., PGM1, PSMB4, LOX) non-informative genes.

To determine the biological function of each layer, we performed the functional enrichment of the associated informative genes. We investigated the predicted functions for BRAC classification as BRCA has the last number of samples (Figure 3B). Since our genes were selected from 14 cancer functional state genesets and 10 KEGG pathways less related to cancer, we used these 24 gene sets for enrichment.

As we can see from Figure 3B, the 4, 3, and 4 pathways were enriched for layers 12, and 3, respectively, (FDR < 0.05). Except one pathway (VALINE_LEUCINE_AND_ISOLEUCINE_DEGRADATION) in the first layer, all other ways are cancer functional states, indicating that T-GEM is able to quickly focus on genes related to cancer even starting from the first layer. Layer 2 and layer 3 focus on specific cancer pathways including metastasis, inflammation, and stemness, which are all highly correlated to breast cancer [38,39,40].

To investigate how the information related to these enriched cancer states is transferred from the input to generate the classification decision, we identified the input Key genes associated with the Query genes in all the significantly enriched pathways for each layer and performed the functional enrichment on these Key genes. Then, for each layer, we linked each enriched pathway with the pathways enriched in Key genes connected to Query genes in that pathway by virtue of Query–Key gene associations (Figure 3B). As shown in Figure 3B, the Key genes from layer 1 are mainly enriched in Invasion, Cell cycle, and DNA Repair, suggesting that only genes in these cancer pathways impact the informative Query genes and thus enriched pathways of layer 1. Interestingly, among these significantly enriched pathways of layer 1, only genes in invasion influence the significantly enriched pathways in layer 2, which are metastasis, inflammation, and stemness. Among these pathways, only invasion and metastasis contribute to the significantly enriched pathways of layer 3, which are invasion, metastasis, inflammation, and stemness, and further contribute to the classification of BRCA. Overall, this investigation reveals that T-GEM attends to more pathways in layers but manages to focus on genes in key cancer pathways in layers 2 and 3 to make decisions.

#### 3.1.3. T-GEM Defines a Regulatory Network That Reveals Marker Genes for Different Cancer Types

Note that the Query–Key gene relationship obtained in the last section, as in Figure 3A, essentially defines a regulatory network that a layer learns with Key genes as the regulators. We extracted the network from the last T-GEM layer for BRCA and LUAD classification using their respective samples and examined the networks (Figure 4).

For BRAC, we first examined the nodes in this regulatory and tested their differential expression between breast cancer patients and normal samples by using the Mann–Whitney U test. Overall, 102/107 genes showed significant differential expression in breast cancer patients compared to normal samples (FDR < 0.05). We further examined the five not-differentially-expressed genes (ABAT, NFKBIA, GUCY1A1, CDH2, TPM2) and found that they do play an important role in breast cancer. For example, ABAT can suppress breast cancer metastasis [41], and NFKBIA’s deletion and low expression are highly related to the unfavorable survival of breast cancer patients [42]. CDH2 is strongly associated with several stem cell-related transcription factors, and it could be the targeted therapy for breast cancer [43]. TPM2 is a potential novel tumor suppressor gene for breast cancer [44]. This result suggests that T-GEM can focus on functional relevant genes even if they show small differential expression. 

Close examination of the network (Figure 4) revealed several hub genes, including GATA3, FOXA1, COMP, IGFBP2, COX7A1, CDH2, VEGFD, AKAP12, and STC2. All of them are highly related to breast cancer, and some of them are known BRCA markers. For example, GATA3 is a well-known associated with favorable breast cancer pathologic features and a promising prognostic marker for breast cancer [45]; FOXA1 is a transcription factor that is predominantly expressed in breast cancer and is considered as breast cancer biomarker [46,47]; COMP is an emerging independent prognostic marker for breast cancer patients [48]. IGFBP2 is directly related to estrogen receptor status, and it is a potential biomarker for the therapeutic response to breast cancer [49]. Finally, STC2 has been shown to suppress breast cancer cell migration and invasion [50].

For LUAD, 183/199 nodes show significant differential expression in LUAD patients compared to normal samples (*U*-test, FDR < 0.05). Similarly, many of the remaining genes that are not differentially expressed (FBN1, COX7A2L, IGFBP2, WNT5B, CDKN1B, KIT, etc.) have been shown to be related to lung cancer. For example, FBN1 plays a crucial role in EMT, and its gene expression is highly correlated with the recurrence-free survival rate for LUAD patients [51]. COX7A2L is important for the assembly of mitochondrial respiratory supercomplexes, and it is related to the regulation of respiratory chain supercomplex assembly and function in lung cancer [52]. A high circulating level of IGFBP2 is highly associated with a poor survival rate, and it is a penitential prognostic biomarker for lung cancer [53].

Similar to BRCA, we observed hub gens NKX2-1, ST6GALNAC1, FOXA1, HSD17B6, AKAP12, DDIT4, ASPN, CEACAM5, MALAT1, GALNT6, and PIPOX (Figure 5), and they also highly associated with LUAD. NKX2-1 controls lung cancer development, and it could be a novel biomarker for lung cancer [54]. AKAP12 acts as a tumor promoter in LUAD and is negatively correlated with patients’ prognosis [55]. A high expression level of DDIT4 associates with poor survival in LUAD patients, and it is a predictor of overall survival [56]. ASPN is found to correlate with LUAD patients’ survival time [57]. HSD17B6 is shown as a potential tumor suppressor in LUAD and a promising prognostic indicator for LUAD patients [58]. CEACAM5 is indicated as a valid clinical biomarker and promising therapeutic target in lung cancer [59]. MALAT1 is a highly conserved nuclear noncoding RNA (ncRNA) and a predictive marker for metastasis development in lung cancer [60]. ST6GALNAC1 can promote lung cancer metastasis, and it is also a novel biomarker for LUAD [61,62]. FOXA1 plays an oncogenic role in lung cancer, and its high expression level is associated with a poor overall survival rate [63]. GALNT6 can promote invasion and metastasis of LUAD, and its expression level is associated with LUAD lymph node metastasis and poor prognosis [64,65]. PIPOX is a major enzyme in the sarcosine metabolism pathway, and it is positively correlated with shorter overall survival [66].

In summary, T-GEM’s attention mechanism models a regulatory network of Query genes by Key genes, and the hub genes of the network of the top layer reveal important marker genes of different cancer types.

### 3.2. T-GEM’ Performance for PBMC Single Cell Cell-Type Prediction and Interpretation Result 

To further examine the generalizability of T-GEM, we applied it to the PBMC single-cell data set for cell type classification. The dataset contains 10 different cell types and 2000 samples for each cell type. Overall, 1000 highly variable genes were selected as the input features. We followed the same model training and selection process and obtained a T-GEM model with one layer and five heads with GeLU as the activation function of the classification layer. 

T-GEM achieved among the best performances, outperforming random forest and decision trees by large margins but was competitive with CNN and SVM (Table 5). We further examined the confusion matrices of SVM and T-GEM separately and observed that both models have high misclassification rates for three T cell subtypes, CD4+ T helper, CD4+/CD25 T Reg, CD4+ /CD45RA+/CD25- Naïve T. However, T-GEM has lower misclassification rates (150/400 and 73/400) for T helper cell and T Reg cell than SVM (157/400, 94/400), although SVM has fewer misclassifications (50/400) for Naïve T cell than T-GEM (82/400). In conclusion, T-GEM achieved competitive performances with SVM or CNN for PBMC cell type classification using scRNA-seq data. However, CNN and SVM are not easy to interpret. We investigate the interpretability of T-GEM next.

#### 3.2.1. T-GEM Learns the Biological Functions That Define Each Cell Types in PBMC

As shown for TCGA cancer type classification, T-GEM makes decisions by focusing on the genes in cancer-related pathways. Therefore, we used the same approach to examine the functions that T-GEM leans. We enriched the informative genes from the first layer in GO Biology Processes. The result shows that most of the informative genes are enriched on the ribosome- and translation-related pathways, including, for example, cytoplasmic translation, peptide biosynthetic process, ribosome biogenesis, ribosome assembly, etc.(the detailed enrichment result is in the Appendix A) Because these cell types are FACS-sorted based on the protein activities of surface markers, but T-GEM’s inputs are gene expression, T-GEM seems to focus more on translation-related processes that translate gene expression to protein activities. We subsequently compared the informative genes with the marker genes for these PBMC cell types defined in the CellMatch database [67] and identified the NK cell markers NKG7 and KLRB1 and the B cell markers CD79B and CD37. Note that many of the marker genes (such as CD4, CD19, etc.) were excluded from the 1000 HVGs selected for our prediction, suggesting that their expression levels are either low or not differentially expressed in the cell types. This result highlights T-GEM’s ability to learn the underlying processes that relate predicted phenotypes with inputs.

#### 3.2.2. The T-GEM’s Regulatory Networks for NK Cell and B Cell 

We next derived the regulatory network of Key-Query genes for the T-GEM layer for the NK cells and B cells (Figure 6 and Figure 7). Out of 123 nodes in this network, 114 showed differential expression in NK cells (FDR < 0.05). For NK cells, there are four hub genes, including MALAT1, CD52, RPLP1, and KLRB1 (Figure 6). KLRB1 is a C-type lectin-like receptor of NK cells and a known NK cell maker. It plays an important role as an inhibitory receptor of NK cells and also works as a co-stimulatory receptor to promote the secretion of IFNγ [68]. MALAT1 is a druggable long non-coding RNA and plays an important role in NK cell-mediated immunity [69].

We next examined the regulatory network for CD19+ B cells (Figure 7). Note that CD19 is not an HVG. Out of 128 nodes in this network, 127 showed differential expression in B cells (FDR < 0.05). CD53 is the only gene that is not differentially expressed. However, CD53 is found to promote B cell development by maintaining IL-&R signaling [70]. We examined the hub genes of the B cell regulatory network, which include HLA-DPB1, RPL35, RPLP0, RPLP1, RPL13, and MALAT1. Interestingly, RPL35, RPLP0, RPLP1, and RPL13 are ribosomal proteins, which could contribute to the enriched translation-related functions of this layer. MALAT1 is also a hub gene for the NK cell network. It has large expression variations across different PBMC cell types and is down-regulated in B cells [71]. HLA-DPB1 is shown to be highly expressed in B cell lines [72] and could serve as an expression marker for B cells. Taken together, these results demonstrate T-GEM’s ability to learn functions and marker genes associated with predicted phenotypes.

## 4. Conclusions

We proposed T-GEM, an interpretable deep-learning architecture inspired by Transformer for gene expression-based phenotype predictions using RNA-seq and scRNA-seq data. In the model architecture, the input gene expression is treated as the token, and a self-attention function attending to markers was proposed to model gene–gene interactions. We comprehensively investigated T-GEM’s performance and learned functions for cancer type classification using TCGA data and immune cell type classification using PBMC scRNA-seq. We show that
T-GEM can provide accurate gene expression-based prediction. We investigated T-GEM for cancer type classification using TCGA data and immune cell type classification using PBMC scRNA-seq. T-GEM has the best performance on both datasets compared with CNN and several classical classifiers.T-GEM learns associated cancer functions. We showed that T-GEM had broad attention in the first layer and paid attention to specific cancer-related genes in layers 2 and 3 for cancer type classification. We also revealed that T-GEM learned cancer-associated pathways at every layer and could concentrate on specific pathways important for predicted phenotypes in layer 3.We extracted the regulatory network of layer 3 and showed that the network hub genes were likely cancer marker genes. We also demonstrated the generalization of these results for immune cell type classification using PBMC scRNA-seq data.

As far as we know, this is the first time such a model has been proposed for modeling gene expression data. Inevitably, there are still many challenges to be addressed. High memory overhead associated with the training of the attention weights limits the input gene dimension. Right now, we have to carefully preselect a subset of genes, which could overlook important genes for T-GEM to model. Furthermore, the self-attention function in Equation (2) is general and allows the integration of prior biological networks such as protein–protein interaction networks to introduce biological meaningful inductive bias into the learning. Research into solutions to these challenges will be our future focus.

## Figures and Tables

**Figure 1 cancers-14-04763-f001:**
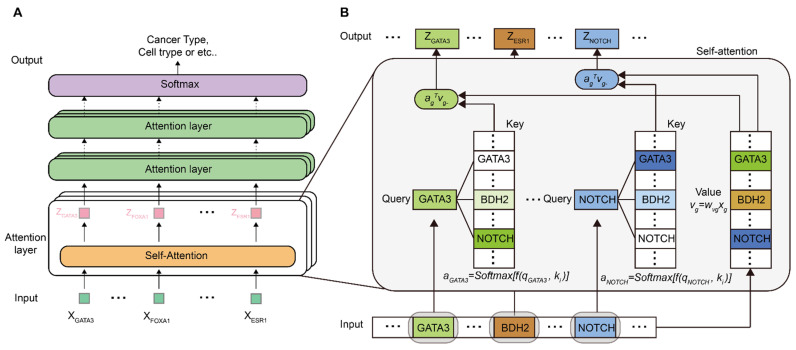
Illustration of the proposed T-GEM model. (**A**) General structure of T-GEM. (**B**) Detailed structure of a head in the first attention layer.

**Figure 2 cancers-14-04763-f002:**
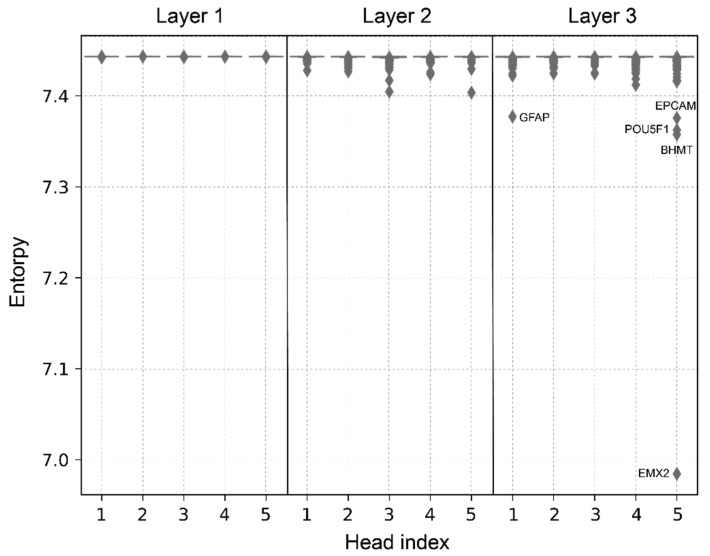
Entropies of each head’s attention weights. Dot plots for different layers are split by black lines. Each dot represents the average entropy value for one query gene.

**Figure 3 cancers-14-04763-f003:**
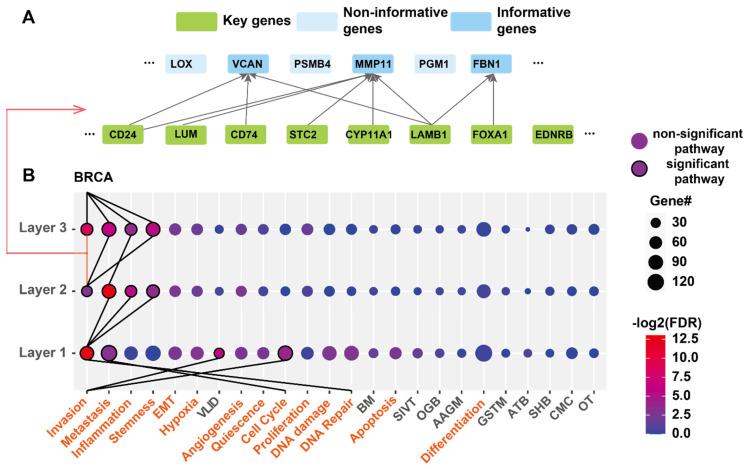
Functions of T-GEM each layer for cancer type classification. A. snapshot of layer 3 after thresholding the weight attribution scores. The links are associated with scores larger than the threshold. Query genes with no links are non-informative genes. (**B**) Enriched functions of each layer for the classification of breast cancer (BRCA). (**A**) link connects an enriched pathway with a pathway in the previous layer if this pathway’s informative gene-associated Key genes are enriched the pathway at the previous layer. The size of the dots represents the number of enriched informative genes in each pathway genesets, and the color shows the enrichment significance (FDR). FDR is negative log2 transferred, red means more significantly enriched, and blue means less enriched. The genesets from cancer states are marked as orange, and genesets from KEGG pathway are black. The pathway is ranked based on the sum of −log2(FDR) for all 3 layers. Abbr. VLID (VALINE_LEUCINE_AND_ISOLEUCINE_DEGRADATION); BM (BUTANOATE_METABOLISM); SIVT(SNARE_INTERACTIONS_IN_VESICULAR_TRANSPORT); OGB (O_GLYCAN_BIOSYNTHESIS); AAGM(ALANINE_ASPARTATE_AND_GLUTAMATE_METABOLISM); GSTM (GLYCINE_SERINE_AND_THREONINE_METABOLISM); ATB(AMINOACYL_TRNA_BIOSYNTHESIS) SHB (STEROID_HORMONE_BIOSYNTHESIS); CMC(CARDIAC_MUSCLE_CONTRACTION); OT(OLFACTORY_TRANSDUCTION).

**Figure 4 cancers-14-04763-f004:**
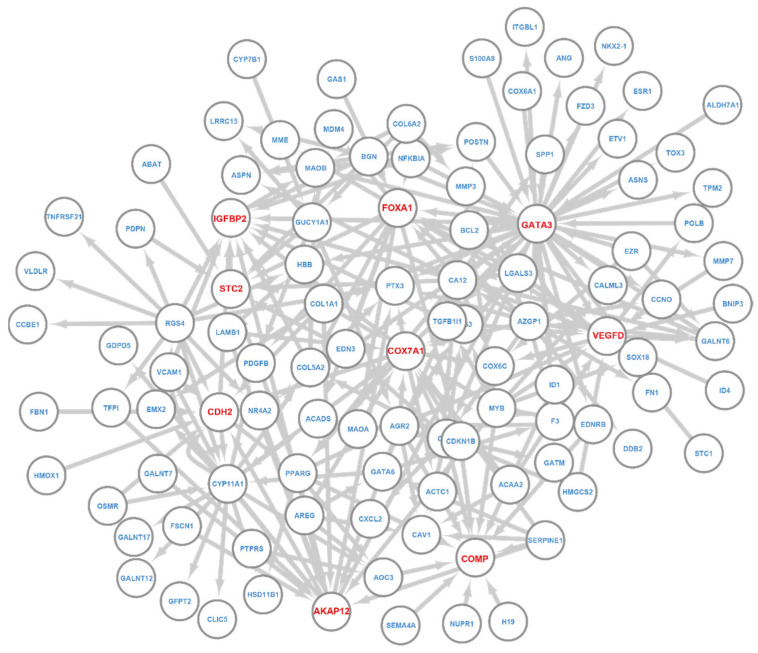
T-GEM regulatory network of the last layer for BRCA. Nodes are informative genes and their associated Key genes. Links go from Key to Query genes. Genes with red color are hub genes.

**Figure 5 cancers-14-04763-f005:**
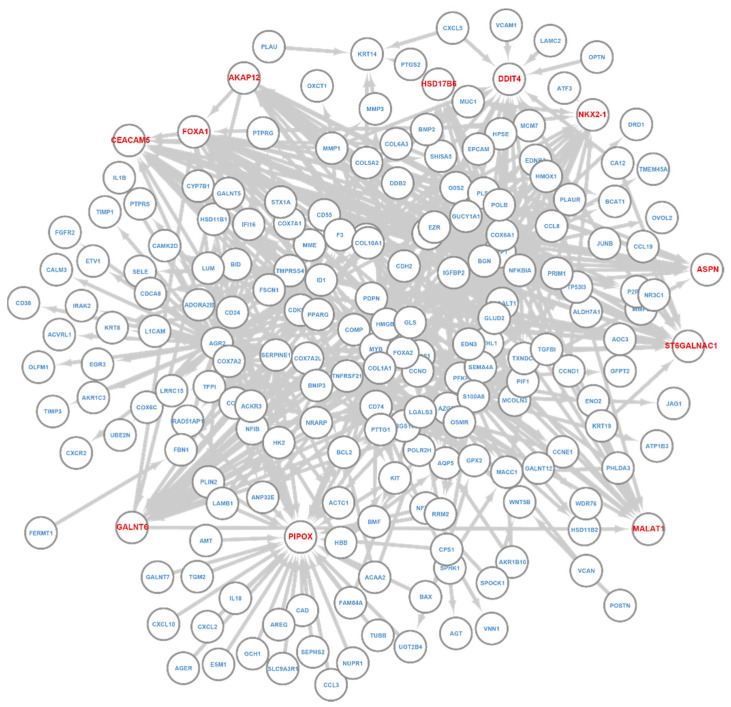
T-GEM regulatory network of the last layer for LUAD. Nodes are informative genes and their associated Key genes. Links go from Key to Query genes. Genes with red color are hub genes.

**Figure 6 cancers-14-04763-f006:**
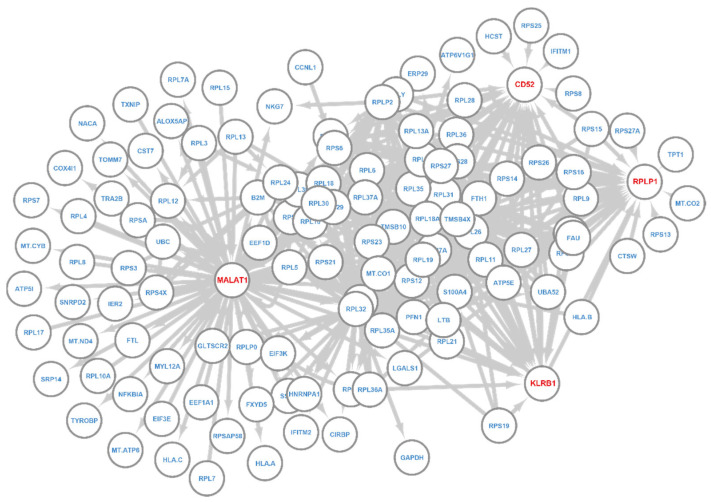
T-GEM regulatory network for NK cell. Nodes are informative genes and their associated Key genes. Links go from Key to Query genes. Genes with red color are hub genes.

**Figure 7 cancers-14-04763-f007:**
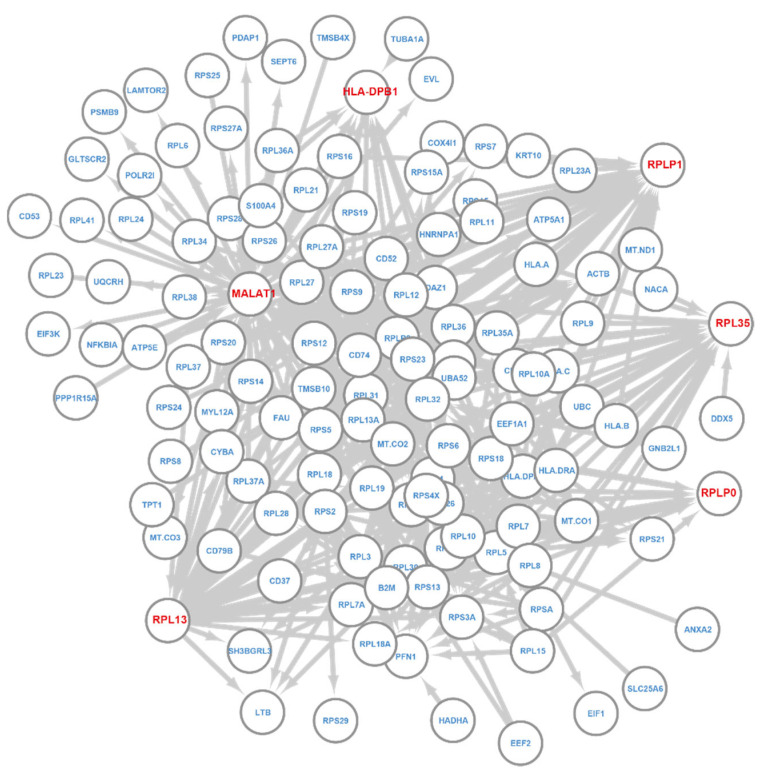
T-GEM regulatory network for B cell. Nodes are informative genes and their associated Key genes. Links go from Key to Query genes. Genes with red color are hub genes.

**Table 1 cancers-14-04763-t001:** Hyperparameter sets of G-TEM.

Hyperparameter	Value of Hyperparameter
The number of layers:	1, 2, 3, 4
the number of heads	1, 2, 3, 4, 5
the activation function of the classification layer	No activation, ReLu, GeLu

**Table 2 cancers-14-04763-t002:** Performances of T-GEM and the benchmark models for TCGA cancer type classification.

	ACC	MCC	AUC
CNN(AUTOKERAS)	94.34%	0.9411	0.9985
SVM	93.21%	0.9292	0.9972
RANDOM FOREST	91.60%	0.9123	0.9970
DECISION TREE	81.80%	0.8097	0.9062
T-GEM	94.92%	0.9469	0.9987

**Table 3 cancers-14-04763-t003:** Accuracy (%) of pruning a T-GEM layer or head for TCGA-based cancer type classification (original accuracy is 94.92%).

	LAYER	HEAD 1	HEAD 2	HEAD 3	HEAD 4	HEAD 5
LAYER 1	57.48	87.19	94.92	95.10	94.92	94.83
LAYER 2	94.83	95.01	94.92	94.61	94.92	94.92
LAYER 3	94.92	94.97	94.92	94.92	94.92	94.97

**Table 4 cancers-14-04763-t004:** Accuracies of each T-GEM layer/head output via SVM.

	LAYER	HEAD 1	HEAD 2	HEAD 3	HEAD 4	HEAD 5
LAYER 1	93.57%	93.48%	91.69%	92.36%	91.91%	93.12%
LAYER 2	93.12%	90.65%	90.88%	93.80%	90.43%	93.57%
LAYER 3	91.24%	93.35%	90.29%	90.61%	89.57%	88.18%

**Table 5 cancers-14-04763-t005:** Performances of T-GEM and benchmark models for cell type classification using PBMC scRNA-seq data.

	ACC	MCC	AUC
CNN(AUTOKERAS)	89.00%	0.8779	0.9945
SVM	90.70%	0.8970	0.9913
RANDOM FOREST	82.53%	0.8062	0.9870
DECISION TREE	74.00%	0.7112	0.8556
T-GEM	90.73%	0.8971	0.9964

## Data Availability

Code and data are available at: https://github.com/TingheZhang/T-GEM (accessed on 1 April 2022).

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
