# Peer review of "Transformer for Gene Expression Modeling (T-GEM): An Interpretable Deep Learning Model for Gene Expression-Based Phenotype Predictions"

_cancers, 2022, doi:10.3390/cancers14194763_

Round 1
Reviewer 1 Report
• How were extreme/outlier values in the data determined and resolved?
• What approaches were used to test the validity of the models? • Which metrics were used in the performance evaluation of the estimates of models/algorithms? • How were the predictive models selected in this study? • Which method(s) was/were used to optimize the hyperparameters of models/algorithms? • How was the most suitable cut-off point determined using the receiver operator characteristic (ROC) curve analysis?Author Response
We greatly appreciate the reviewers’ comments and thank them for their time and valuable suggestions. The following revisions have been made in accordance with the suggestions from the reviewers. The revisions are highlighted in the manuscript in blue ink.
Reviewer 1
- 1 How were extreme/outlier values in the data determined and resolved?
Response 1: We examined the distribution of the log-transformed and normalized 1,708 genes and did not observe any extreme/outlier values.
- 2 What approaches were used to test the validity of the models?
Response 2: We tested our model in two aspects. First, we validated if our model can classify cancer types by comparing our model with other benchmark methods (Table 2). Second, we validated if our model can learn the underlying functions and penitential marker genes by examining if the functions that each layer learned are cancer related functions and the hub genes of the regulatory networks in each layer agree with experimentally validated marker genes (Figure 4 and Figure 5).
- 3 Which metrics were used in the performance evaluation of the estimates of models/algorithms?
Response 3: In this paper, we used 3 performance metrics, i.e. accuracy (ACC), Matthews correlation coefficient (MCC), and the area under the ROC curve(AUC). The detailed information can be found in section 2.5.
- 4 How were the predictive models selected in this study?
Response 4: In this paper, we trained the model for 50 epochs and evaluated the model performance on the validation dataset for each epoch. The predictive model is selected from the epoch with the highest accuracy.
- 5 Which method(s) was/were used to optimize the hyperparameters of models/algorithms?
Response 5: Our model has 3 hyperparameters, the number of layers, the number of heads and the activation function of the classification layers (Table 1). We used the validation dataset to select the best combination of hyperparameters that produced the highest accuracy.
- 6 How was the most suitable cut-off point determined using the receiver operator characteristic (ROC) curve analysis?
Response 6: Thanks for comment. We presented the area under the ROC curve (AUC) instead of applying the cut-off to the ROC curve. We calculated all the performance metrics by using package scikit-learn[1]
- Pedregosa, F., et al., Scikit-learn: Machine learning in Python. the Journal of machine Learning research, 2011. 12: p. 2825-2830.
Reviewer 2 Report
The manuscript is very interesting and very well written. However, some minor mistakes should be corrected:
1. Throughout the paper check for grammar and spelling errors.
a. In the abstract correct the following:
i. …challenge the computer vision inspired… to challenge the computer vision-inspired…
ii. …in developing interpretable DL models… to …in developing an interpretable DL model…
iii. …phenotype related… to …phenotype-related
2. If an abbreviation appears for the first time in the manuscript text, then the full term should appear, and the abbreviation is placed in parentheses. Every other time the term is used in the text, then only use abbreviation.
a. For example in the abstract: …design of popular DL models such as CNN… In this case the DL and CNN appear for the first time so it should be written as. …design of popular Deep learning (DL) models such as Convolutional Neural Network (CNN)…
3. Put one space between the text and the reference. For example …the associated functions[13, 14]… change to …the associated functions [13,14].
4. The Introduction section should be expanded. Provide more references that describes the previous research. Also, at the end of the introduction clearly and explicitly state the hypothesis of your research, preferably in bullet form. After that put one paragraph that clearly and shortly describes the further sections of your paper (one to maximum of two sentences for each section of the paper).
5. After the title “Materials and Methods” shortly describe what will be described in this section, the same goes for the subsection “Data collection and preprocessing” and of course any other section/subsection title.
6. Center the equation number 2
7. Improve the quality of Figure 1, higher dpi if possible, and enlarge the figure if possible.
8. Put the gird on Figure 2, enlarge the figure and improve its quality (higher dpi)
9. Improve the quality (higher dpi) of Figure 3, enlarge the Figure if possible.
10. Improve the quality (higher dpi) of Figure 4,5,6,7 , enlarge the Figure if possible. At this form Figure 4 is not readable.
11. Change the title Results to Results and Discussion since you clearly discussing the results in the Results section
12. Rewrite the conclusion, preferably in bullet form to clearly indicate the conclusions extracted from the results and discussion section. The conclusion section should also provide answers to hypotheses defined in the Introduction section.
Author Response
We greatly appreciate the reviewers’ comments and thank them for their time and valuable suggestions. The following revisions have been made in accordance with the suggestions from the reviewers. The revisions are highlighted in the manuscript in blue ink.
- Throughout the paper check for grammar and spelling errors.
In the abstract correct the following:
- …challenge the computer vision inspired… to challenge the computer vision-inspired…
- …in developing interpretable DL models… to …in developing an interpretable DL model…
- …phenotype related… to …phenotype-related
Response 1: We thank the reviewer for the careful reading of the manuscript We have corrected all the errors.
- If an abbreviation appears for the first time in the manuscript text, then the full term should appear, and the abbreviation is placed in parentheses. Every other time the term is used in the text, then only use abbreviation.
For example in the abstract: …design of popular DL models such as CNN… In this case the DL and CNN appear for the first time so it should be written as. …design of popular Deep learning (DL) models such as Convolutional Neural Network (CNN)…
Response 2: Thank you for your comments. We have revised it as suggested.
- Put one space between the text and the reference. For example …the associated functions[13, 14]… change to …the associated functions [13,14].
Response 3: Thank you for your suggestion. We have added space between text and reference.
- The Introduction section should be expanded. Provide more references that describes the previous research. Also, at the end of the introduction clearly and explicitly state the hypothesis of your research, preferably in bullet form. After that put one paragraph that clearly and shortly describes the further sections of your paper (one to maximum of two sentences for each section of the paper).
Response 4: We have expanded our introduction to include more discussion of previous work. We have also included a discussion on the hypothesis and a paragraph to summarize each section of the paper.
- After the title “Materials and Methods” shortly describe what will be described in this section, the same goes for the subsection “Data collection and preprocessing” and of course any other section/subsection title.
Response 5: We have updated section 2 and section 3 and added descriptions as suggested by the reviewer. We also merged subsections 3.1 and 3.1.1.
- Center the equation number 2
Response 6: We have centered equation 2.
- Improve the quality of Figure 1, higher dpi if possible, and enlarge the figure if possible.
Response 7: We used 300ppi previously, but MS Word may decrease the figure quality. We included raw figure (pdf version) in the supplementary files and changed the figures in the paper to 600ppi. We will submit the high-quality figure if the paper is accepted.
- Put the gird on Figure 2, enlarge the figure and improve its quality (higher dpi)
Response 8: We have added the gird on Figure 2 and changed it resolution to 600ppi.
- Improve the quality (higher dpi) of Figure 3, enlarge the Figure if possible.
Response 9: Done.
- Improve the quality (higher dpi) of Figure 4,5,6,7 , enlarge the Figure if possible. At this form Figure 4 is not readable.
Response 10: Done.
- Change the title Results to Results and Discussion since you clearly discussing the results in the Results section
Response 11: We have revised them as suggested.
- Rewrite the conclusion, preferably in bullet form to clearly indicate the conclusions extracted from the results and discussion section. The conclusion section should also provide answers to hypotheses defined in the Introduction section.
Response 12: We have revised the Conclusion section according to the reviewer’s suggestion.
